# A Systematic Review Protocol for the Effectiveness of Psycho-Educational Intervention Programmes in Addressing the Psychological Risk Factors Associated with Non-Communicable Diseases among Adolescents

**DOI:** 10.3390/ijerph20156467

**Published:** 2023-07-28

**Authors:** Nokwanda P. Bokolo, Rudolph Leon Van Niekerk, Verona Mathews, Lloyd Leach

**Affiliations:** 1School of Public Health, University of the Western Cape, Bellville 7535, South Africa; 2Department of Psychology, Faculty of Social Sciences and Humanities, University of Fort Hare, Private Bag X9083, 50 Church Street, East London 5201, South Africa; 3Department of Sport, Recreation and Exercise Science, University of the Western Cape, Bellville 7535, South Africa

**Keywords:** adolescents, intervention, psycho-educational programmes, non-communicable diseases, risk factors, psychological health

## Abstract

The psychological health concerns and risk factors associated with non-communicable disease among adolescents have been prioritized on the agenda of international health institutions globally. The aims of this systematic review are to determine the various types of psycho-educational intervention programmes developed to address the psychological risk factors associated with non-communicable diseases among adolescents, and to assess the effectiveness of the psycho-educational intervention programmes. The systematic review will include case–control and randomized controlled studies. The review will be conducted using the following electronic databases, PubMed, CINAHL, Science Direct, Cochrane Library, SCOPUS, and ERIC, as well as the grey literature for the thesis repository from 2012 to 2022. The key search terms will include intervention programme, adolescents (aged 10–19 years), psychological risk factors and non-communicable diseases. The studies identified by the search strategy will be downloaded into Mendeley and exported to Covidence software for screening, quality assessment and data extraction. The quality assessment tool that will be utilized is the Joanna Briggs Institute critical appraisal checklists to ensure relevance and quality of the articles. This systematic review will use two types of data analysis: narrative synthesis of qualitative studies and meta-analysis of quantitative studies. The findings from this systematic review will provide evidence-based tools for the management of psychological risk factors associated with non-communicable diseases, as well as present key insights for future intervention programmes on the management of psychological risk factors associated with NCDs among adolescents.

## 1. Introduction

Adolescence is an important phase in human development and it refers to individuals between the ages of 10 and 19 years [1]. It is a unique stage and an important time for laying the foundations of good health [2]. However, during this stage, adolescents can establish patterns of behaviour related to diet, physical activity, substance use and sexual activity that often put their health at risk [3]. It is also an exceptional and formative stage, where physical, social and emotional changes render adolescents vulnerable to psychological health conditions that can negatively impact their development [4]. Additionally, uninformed and abusive parents often present major concerns for adolescents’ overall health and well-being, as this may result in several negative psychological, behavioural and physical health outcomes [5].

The psychological health and well-being of adolescents are continuous global public health concerns [6]. Although there were concerns about the psychological [4] health of youth prior to the COVID-19 pandemic, more recently among the world’s 1.2 billion adolescents, psychological health conditions account for approximately 16% of the global burden of disease and injury [4], while 85% of premature deaths are estimated to occur in low- and middle-income countries [7]. The COVID-19 pandemic has been reported to have severely impacted the well-being of young people and has put them at an increased risk of suicide, substance use and other psychological health problems [8]. A steadily growing body of literature shows that the COVID-19 pandemic has had a deleterious impact on youth, including their psychological health, with an estimated 38% of children and adolescents experiencing psychological health conditions, i.e., anxiety [9,10]. Recent evidence suggests that half of all lifetime psychological health conditions start during adolescence, yet the majority of these will go unrecognized and untreated [11]. Various factors hinder the effective treatment of adolescent psychological health, such as the lack of proper healthcare resources, the lack of effective communication between parents and healthcare practitioners, the lack of policies aimed at child and adolescent health, and the exposure of adolescents to psychological risk factors, particularly in disadvantaged communities [6].

Psychological risk factors, such as psychological distress, and mental and emotional stress have contributed to non-communicable diseases (NCDs) [12]. Non-communicable diseases are defined as medical conditions that are non-infectious and cannot be passed from person to person [13]. They are known to result in long-term health consequences and often create a need for prolonged treatment and care [14]. The risk factors for NCDs are often stated as either behavioural [15] or psychosocial [16]. Recent research highlighted a strong relationship between psychological distress and various NCD risk factors among adolescents [17].

The psychological health of young people is often affected by their living conditions, the prevailing social stigma and/or discrimination and a lack of access to quality social support and healthcare services [4]. In order to cope (psychologically) with their challenging and difficult circumstances, adolescents often adopt risky behaviours that include unhealthy diets, smoking, and drugs or alcohol use [18]. Additionally, social and environmental factors, such as the influence of parents and peers, community norms and the mass media, help to explain the reasons why adolescents are at risk [19]. The risky behaviours that adolescents and young people engage in become a serious concern from a health perspective [20]. Regardless of the best efforts being made to educate the youth about the harmful effects of risky behaviours, research evidence shows that South African adolescents are engaging in behaviours at an alarming rate without too much concern for the consequences of their actions [21]. In this regard, psychological risk factors appear to be the precursors and result in the development of various behavioural risk factors [22]. Additionally, adolescents face developmental challenges such as sexual development and identity and social issues, i.e., self-identity, and peer pressure [19]. These risk factors are also known to result in the development of NCDs in the long-term [23]. The most common NCDs among adolescents include obesity, diabetes, high blood pressure, cancer, raised cholesterol and respiratory diseases [24].

Non-communicable diseases and their risk factors have become a major public health burden worldwide [25]. The NCDs among adolescents are of increasing concern as well, for societies and national governments, as well as globally, due to the significant impact they have on morbidity and mortality rates [13]. Approximately 71% of NCD-related premature deaths worldwide and over 85% of NCD-related premature deaths in low- and middle-income countries occur during adulthood [24]. These are preventable deaths that are the result of modifiable health-related behaviours that are initiated in childhood and adolescence [26]. These adverse health-related behaviours or deleterious conditions include overweight and obesity, physical inactivity, substance use and poor nutrition [13]. It is estimated that 10% of young people smoke, 11.7% are heavy alcohol drinkers and 81% of adolescents have an inadequate amount of physical activity, thus disposing them to the development of NCDs [14]. Many NCD-related risk factors are initiated or reinforced during adolescence, and have a significant impact on children and adolescents across the life-course, which makes it a critical period for applying appropriate interventions in order to prevent future illness and/or disease [24]. However, the existing reviews with regard to the appropriate interventions that can be implemented during early childhood and adolescence are scarce, and reviews have mainly focused on interventions for NCDs in adults [27].

A range of psycho-educational interventions for psychological health promotion have been developed for schools in the last decade with variable degrees of success [28]. However, within these interventions, it has been emphasized that the modes of delivery and the nature of the interventions were particularly important for the overall success of the outcome variables [29]. Equally important, the interventions activities need to appeal to young people, by including mobile devices with digital tools, mobile applications, robotics, social media and the internet [28]. For this review, the focus is on the application of psycho-educational intervention strategies that are known to provide both disease-specific information, i.e., early recognition and management of relapse symptoms, as well as more general information, i.e., promotion of a healthy lifestyle [9]. More especially, the psycho-educational intervention has been defined as the improvement in knowledge in subject-specific areas that serve the goals of treatment and rehabilitation [30]. Additionally, the psycho-educational interventions included information on how to explain to family members certain aspects of healthcare, such as living with an illness so that they could understand the effect of the illness and, thereby, assist the patient and healthcare providers in the treatment regimen [9]. There is a growing interest in psycho-educational interventions in order to provide accurate information about health issues and to enhance and develop self-management skills [31]. However, in the existing literature, there has been a lack of research and engagement, specifically with the psychological risk factors associated with NCDs [32].

The WHO guidelines on the promotion of psychological health and preventive strategies for adolescents have focused on providing information on adolescent psychological health and preventing psychological health conditions, such as self-harm, substance use and other high-risk behaviours [4]. The need to focus on the psychological health of adolescents has gained increasing attention and recognition, as the global community looks to achieve the Sustainable Development Goals (SDGs) that were adopted by the United Nations in 2015 [33]. It is vital to address the main threats to Public Health in order to achieve the SDG targets [34]. These threats include the WHO mental health Gap Action Programme intervention guide by providing global, evidence-based recommendations on promotive and preventive psychological health interventions for adolescents [4]. Invariably, the NCDs threaten progress towards the 2030 Agenda for Sustainable Development, which is focused on reducing premature deaths from NCDs by one-third by 2030 [4]. This initiative aims to achieve universal health coverage through the promotion of prevention and the treatment of physical, psychological and social health challenges, and social well-being [24].

Promoting psychological well-being and preventing adolescents from adverse experiences and risk factors, which may negatively impact their potential to thrive and are critical for their well-being during adolescence and for their physical and psychological health in adulthood [4]. Early intervention strategies that are focused on promoting health and preventing NCDs may produce the greatest impact on adolescents’ health and well-being [26]. These interventions are expected to increase awareness by equipping the adolescents with specific knowledge and skills and, thus, facilitate a change in their risky behaviours [35]. The paradigm shift in the conceptualization of NCDs and risk factors is extremely important in developing effective interventions and policies to prevent and alleviate the increasing NCD-burden among adolescents [24]. Therefore, from a public health standpoint, investments that emphasize using multi-level and cross-sectoral interventions and policies to address the diverse risk factors for NCDs must be prioritized by countries globally in order to achieve improvements in NCD-control among adolescents [25].

In 2020, the WHO continued its commitment to support and promote the psychological health and well-being of adolescents through the Sustainable Development Goals [4]. It recommended the period of adolescence as one of the optimal windows of opportunity for intervention, given the neuroplasticity evident in adolescence and the opportunity to step in at a time when the majority of psychological health conditions and risky behaviours have their onset [35]. Particular attention is given to adolescents who are at increased risk for psychological disorders or self-harm, and adolescents who present with early signs and/or symptoms of emotional and/or behavioural problems [4]. It is more effective to address multiple interrelated health outcomes and associated behaviours through integrated policies and programmes, including interventions [36].

The psychological health concerns and risk factors associated with NCDs among adolescents have been prioritized on the agenda of international health institutions globally, and this has been highlighted as a continuous global public health concern [24]. The World Health Organization supports and promotes interventions to halt this public health concern [4]. To date, there is a lack of evidence on psycho-educational intervention programmes that address the psychological risk factors associated with NCDs among adolescents. This concern for public health raises the need for effective psycho-educational intervention programmes among adolescents aimed at addressing and reducing the psychological risks factors associated with NCDs. Therefore, this review intends to determine the various types of psycho-educational intervention programmes that have been developed to address the psychological risk factors associated with NCDs among adolescents. Also, this review will assess the effectiveness of the various psycho-educational intervention programmes used to address the psychological risk factors associated with NCDs among adolescents.

This systematic review arose from the growing psychological health burden faced globally [37], as well as the risk factors for NCDs among adolescents, that is steadily increasing the global public health challenge and the impact it has on society, especially on those affected and their families [24]. The interventions to manage NCDs among adolescents are insufficient and are targeted primarily at the adult population [38]. Therefore, health researchers and policy experts need to steer evidence-based research to address NCD risk factors and the increasing exposure to NCDs due to the adoption of risky behaviours [39]. The current literature shows that there is still a need for a stronger and broader evidence base in the field of psychological health promotion, which should focus on both universal healthcare and targeted approaches locally in order to fully address the psychological health conditions in our young populations [28]. This study hopes to inspire researchers to design the appropriate psycho-educational intervention programmes to improve the education and prevention of psychological risk factors for NCDs among adolescents, thus facilitating an integrated approach to improve adolescent health.

The objectives of the present systematic review are to determine the various types of psycho-educational intervention programmes that have been developed to address the psychological risk factors associated with non-communicable diseases among adolescents, and to assess the effectiveness of the psycho-educational intervention programmes. In line with the objectives of this systematic review, the PICO (Population, Intervention, Comparison and Outcomes) method is utilized to develop the research questions and to identify gaps in the literature. The research questions for this review are

Which types of psycho-educational intervention programmes have been used to address the psychological risk factors associated with NCDs among adolescents?Which types of psycho-educational intervention programmes were shown to be effective in addressing the psychological risk factors associated with NCDs among adolescents?How effective was each type of the psycho-educational intervention programme in addressing the psychological risk factors associated with NCDs among adolescents?

## 2. Materials and Methods

This systematic review has been registered with the International Prospective Register of Systematic Reviews (Prospero) with the registration number: CRD42021259645. The ethics approval for this systematic review was granted by the Biomedical Research Ethics Committee at the University of the Western Cape (ethics reference number: BM22/6/27). This systematic review protocol has been prepared according to the preferred reporting items for systematic reviews and meta-analyses (PRISMA) 2020 framework [40] in Figure 1 below (see Appendix A). Additionally, the seven steps described by Eggar et al., (2001) will be used to guide the systematic review process. These steps are as follows: (i) formulate the review question; (ii) define the inclusion and exclusion criteria; (iii) develop a search strategy; (iv) select appropriate studies; (v) assess the quality of the selected studies; (vi) extract data from the studies; and (vii) analyse or synthesize the data [41].

### 2.1. Eligibility Criteria

#### 2.1.1. Types of Studies

A detailed and systematic literature search will be conducted to identify the study designs, which include case–control studies and randomized controlled trials (RCTs). These studies will assist in identifying the current knowledge, gaps, and opportunities in the research. In particular, the randomized controlled trials will be prioritised, since they are known to reduce bias and provide a rigorous tool to examine cause–effect relationships between an intervention and outcome [42]. Also, the studies included in the review will help to provide insights concerning the recommendations for policy, action, and transdisciplinary research.

#### 2.1.2. Types of Participants

This review will consider studies conducted on adolescents aged 10–19 years following WHO’s definition of adolescents [4], as those with psychological risk factors or exposed to psychological risk factors associated with NCDs, and who have participated in psycho-educational interventions. This stage is considered as one of the ideal timeframes for intervention, as the opportunity to step in at a time when the majority of psychological health conditions and risky behaviours have their onset [4].

#### 2.1.3. Types of Interventions

In the context of this review, interventions are regarded as suitable evidence-based research to address the problem or burden of psychological risk factors associated with NCDs, and present possible strategies for developing guidelines for best practice. This review will specifically consider studies conducted on psycho-educational intervention programmes or interventions involving an active component to reduce or address the psychological risk factors associated with NCDs among adolescents.

The interventions should consider the broad definition of psychological risk factors (psychological distress, perceived stress and languishing) [11] and behavioural risk factors for NCDs (over-eating, tobacco use, harmful use of alcohol, physical inactivity, etc.) [18]. Interventions will be classified as follows: educational, psycho-educational, cognitive behaviour therapy, social contact and video-based or text-reading education [9]. The interventions considered may be delivered individually or in groups, at home, in the community (e.g., schools) or in health facilities, and using face-to-face, online, or blended/hybrid approaches. The interventions may also be delivered through didactic lectures, photographic images of billboard messages, short educational messages, video messages, structured courses and workshops for students, and through the distribution of booklets and slideshows [43].

There will be no exclusion based on duration of the intervention, length of follow-up, mode of intervention delivery or format of the intervention. The interventions reviewed may be delivered by laypersons, educators, health personnel or specialists in health psychology, among others.

#### 2.1.4. Types of Comparator(s)/Control

The control groups will continue with their daily routine without receiving the intervention. Other types of comparators that have been used in these studies involved treatment such as no contact.

The literature reveals that other interventions have contributed to the prevention, reduction and management of psychological health in adolescents, as this review will consider studies that compare the psycho-educational interventions to other alternative non-pharmacological interventions, i.e., play therapy, mindfulness and psychodynamic [9].

#### 2.1.5. Types of Outcome Measures

The following outcome measures are considered from the studies that meet the inclusion criteria for this review:

Primary outcome measures for this study are mental/psychological state (including self-control, knowledge, and attitudes), quality of life, global state, and behaviours.

Secondary outcome measures are physical, physiological, and clinical biomarkers of NCDs.

### 2.2. Inclusion and Exclusion Criteria

The inclusion and exclusion criteria indicated in Table 1 below:

#### 2.2.1. Context

Eligible studies on intervention programmes for psychological health and/or development of NCDs.Studies published in peer-reviewed journals, including case–control studies, and randomized controlled trials (RCTs).Studies available as full-text publications (open-access or subscription) in English. Titles, abstracts, and texts will be screened against the inclusion criteria by the reviewers.

#### 2.2.2. Exclusion Criteria

Studies will be excluded if they

(a)Are non-English and non-peer reviewed.(b)Do not contain adolescents with psychological risk factors and/or psychological challenges and/or NCDs and/or intervention programmes.(c)The population is generally considered to be healthy, from a psychological standpoint, and have adolescents included in the sample.

### 2.3. Search Strategy

Searches will be conducted for the literature over the past ten years (2012–2022) to account for recent developments in best-practices during the last decade, following the WHO’s call for global action and the WHO’s Collaborating Centre for Health Promotion Research in 2013. A systematic and comprehensive electronic database search will be performed using the Cumulative Index of Nursing and Allied Health Literature (CINAHL), the Educational Resource Information Centre (ERIC), PubMed, SCOPUS, Science Direct and Cochrane Library, as well as the grey literature for the thesis repository, as these are a combination of human-curated databases and search engines on the internet. These databases were chosen considering the multidisciplinary and global nature of the review under investigation, with the goal of covering the literature in the public health and social science domains [44].

The key search terms will include intervention programme, adolescents (aged 10–19 years), psychological risk factors and non-communicable diseases.

Search terms: (“adolescents OR teenagers OR young people”) AND (“psychological risk factors” OR “mental health risk factors”) AND (“NCDs risk factors” OR “NCD development”) AND (“psycho-educational intervention programme” OR “Intervention programme”). In order to conduct an exhaustive and comprehensive search, the review team will expand some of the keywords, as indicated in the Table 2. This expansion will provide broader knowledge and understanding for the review topic. This will also ensure the retrieval of all relevant results.

### 2.4. Study Screening and Selection

The primary search for peer-reviewed relevant articles, including importing and exporting of citations, will be performed by the principal researcher (NB) and peer reviewer (AK). All the abstracts and titles of the selected studies will be downloaded and saved into Mendeley. The downloaded studies will be exported to Covidence software for managing and streamlining the review and removing the duplicates of studies and other studies identified as inappropriate according to the criteria of this review. The title and abstract screening will be conducted in Covidence by the reviewer, peer reviewer and co-supervisors (VM, LvN), which will be checked against the inclusion and exclusion criteria. Disputes between the reviewers will be resolved by the supervisor (LL). For reporting the systematic review, the preferred reporting items for systematic reviews and meta-analyses (PRISMA) framework will be used.

In addition, part of screening the full-text articles will entail determining the relevance of the articles to include in the systematic review. The reasons for excluding articles will be recorded, and any conflicts between reviewers will be resolved by the supervisor (LL).

### 2.5. Data Extraction

A data abstraction form will be developed in Microsoft Office Excel 2010 by the team (NB, AK, VM, LvN). The data abstraction tool will be piloted on a random sample of five articles and modified as per the feedback from the team. Thereafter, data extraction will be performed to extract the information required to respond to the research questions of the systematic review. NB will take full responsibility for all aspects of the review and will be responsible for extracting the data, verifying the data, analysis, grading, and writing up the review. LL will also be responsible for verifying the data and providing general guidance in conducting the review.

The data extraction for determining the effectiveness of interventions will be carried out in the following areas (i) inclusion and exclusion criteria for participation where applicable; (ii) study description (authors, aims, objectives, ethical issues); (iii) trial details (details of location, population, intervention and control used); (iv) intervention details (allocation methods, theoretical basis, content of intervention, delivery, duration, targeted setting); (v) outcome measures used and results with sufficient information; targeted risk factors, type of settings-based health promotion.

The quality assessment of the selected articles will employ the Joanna Briggs Institute (JBI) (see Appendix A) critical appraisal checklist to ensure relevance and quality of the articles and to assess bias. The JBI is determined to improving health outcomes in communities globally by promoting and supporting the use of the best available evidence to inform decisions made at the point of care [45]. JBI is known to ensure that the research evidence we seek to synthesize, transfer and implement is culturally inclusive and relevant across the diversity of healthcare internationally [45]. This produces decision making that considers the feasibility, appropriateness, meaningfulness, and effectiveness of healthcare practice, i.e., intervention programmes [9].

### 2.6. Data Synthesis and Meta-Analysis

This systematic review will involve two types of analyses: narrative synthesis of findings from multiple studies that relies primarily on the use of words and text to summarize and explain the findings of the research, and a meta-analysis of experimental studies. A narrative synthesis of text-based studies will be used to address objective one of the systematic reviews, i.e., to identify similarities and differences between the studies and for obtaining a broad perspective on the review. Also, a narrative synthesis is known to use a textual rather than a statistical approach for analysing results and drawing conclusions [46]. In addition, a meta-analysis of quantitative studies will be used to address objective two of the review, i.e., to describe the interventions and measure the outcomes, specifically knowledge, attitudes and behaviours regarding psychological risk factors associated with NCDs. This will include reporting the mean differences between the change in intervention and control group with respect to quantitative data outcomes. Where there is no change per group, the end values where randomization was successful will be used. Studies will be grouped according to their similarities regarding the methods or techniques used to address the psychological risk factors associated with NCDs among adolescents.

### 2.7. Author Contributions

NB and LL drafted the first draft of the protocol and LvN and VM shared their expertise on drafting the manuscript. All authors read, provided feedback, and approved the final version of the manuscript.

## 3. Strengths and Limitations

A limitation of the systematic review is that the search focus will be limited to articles published in English only and may exclude pertinent studies written in other languages. This review will also be limited to adolescents between the ages of 10 and 19 years, which will omit the literature from early childhood and adulthood. The review will be limited to a period of ten years, i.e., from 2012 to 2022. A strength of the review will be identifying best practices on the effectiveness of psycho-educational intervention programmes for adolescents aimed at addressing and reducing the psychological risk factors associated with NCDs.

## 4. Conclusions

The results of this systematic review will provide evidence-based tools for the management of psychological risk factors associated with non-communicable diseases, as well as present key insights for future developments of intervention programmes on psychological risk factors associated with NCDs among adolescents. The results can be used to inform future research projects and guide policymakers and health professionals about the enhancement of adolescents’ knowledge, attitudes and behaviours concerning the psychological risk factors associated with NCDs. The outcomes of this study can help to complement the scarce knowledge on the psychological well-being and psychological risk factors associated with NCDs among adolescents, especially in South Africa. The findings may also be useful for schools and organizations to understand the dynamics of adolescent maladaptive behaviour and, thereby, help to create an awareness of NCD development, and promote the psychological well-being of adolescents from impoverished backgrounds and marginalised communities. This review holds significant practical implications for the field of adolescent health and well-being by identifying and examining the different types of psycho-educational intervention programmes specifically designed to target the psychological risk factors linked to NCDs in adolescents. Also, an assessment of the effectiveness of these psycho-educational interventions will provide valuable insights into their efficacy in addressing and mitigating the psychological risk factors associated with NCDs in adolescents. Ultimately, the findings from this review can inform the development and implementation of evidence-based intervention programmes aimed at promoting the psychological well-being of adolescents and reducing their vulnerability to NCDs.

## Figures and Tables

**Figure 1 ijerph-20-06467-f001:**
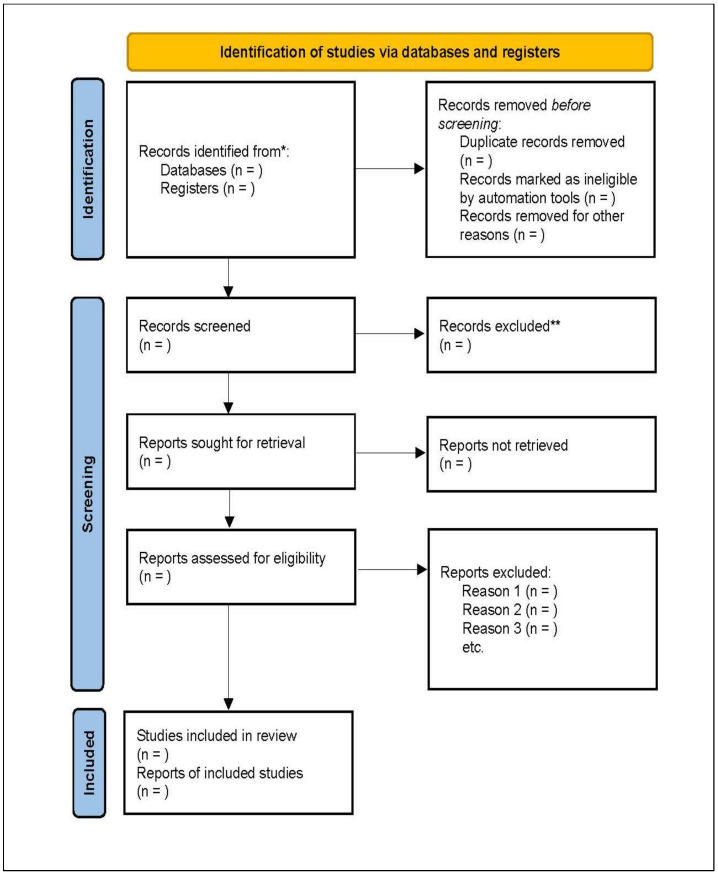
PRISMA 2020 flow diagram. * Indicates that the number of records identified from each of the six databases or registers searched will be reported, as well the total number across all of the databases/registers. ** Indicates the number of records that will be excluded by human means and by automation tools. n indicates the number of records that will be reported from each of the databases or registers searched.

**Table 1 ijerph-20-06467-t001:** Inclusion and Exclusion criteria.

PICO	Inclusion Criteria	Exclusion Criteria
Population	Studies on adolescents (aged 10–19 years) with psychological challenges OR exposed to psychological risk factors associated with NCDs.	Studies that do not contain data on adolescents’ psychological challenges or risk factors associated with NCDs. Studies on heterogeneous populations or without a definite age-range.
Intervention	Studies on psycho-educational intervention programmes OR Interventions involving an active component to reduce/address the psychological risks factors associated with NCDs among adolescents.	Studies that do not contain intervention programmes or prevention of NCDs among adolescents.
Comparison/Control	Studies that compare the experimental groups that participate in the intervention programme, while the control groups continue with their daily routines.	Studies with unclear comparison of the experimental groups that participate in the intervention programme.
Outcomes	Studies reporting on any objective’s measures of health, i.e., increased self-control of phycological risk factors, including knowledge, attitudes and behaviours of adolescents concerning the psychological risk factors associated with NCDs. Studies reporting on measures of physical, physiological, and clinical biomarkers in adolescents regarding the psychological risk factors associated with NCDs.	Studies that do not include measures in knowledge, behaviours and/or attitudes of adolescents regarding psychological risk factors associated with NCDs. Studies that do not include measures in changes in physical, psychological, and clinical biomarkers in adolescents regarding the psychological risk factors associated with NCDs.

**Table 2 ijerph-20-06467-t002:** Search Terms and Alternatives.

Terms	Alternatives
Adolescents	Teenagers, youth, young adults
Psychological risk factors, i.e., perceived stress, psychological distress, languishing, emotional stress, mental stress	Mental health risk factors
Non-communicable diseases (NCDs), i.e., overweight, diabetes, hypertension, hypercholesterolemia, and respiratory diseases	NCD risk factors or NCD development, i.e., tobacco use, physical inactivity, unhealthy diet, and harmful use of alcohol
Psycho-educational intervention OR programme OR intervention programme	Intervention programme

## Data Availability

Not applicable.

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
