# Peer review of "A Systematic Review Protocol for the Effectiveness of Psycho-Educational Intervention Programmes in Addressing the Psychological Risk Factors Associated with Non-Communicable Diseases among Adolescents"

_ijerph, 2023, doi:10.3390/ijerph20156467_

Round 1

Reviewer 1 Report (Previous Reviewer 1)

Authors have included the improvements requested by the reviewers and I believe that this article can be accepted.

Author Response

Dear Reviewer

Thank you for your constructive comments that assisted in improving the manuscript.

Warm regards

Reviewer 2 Report (Previous Reviewer 2)

Dear authors,

This protocol is useful and important

-The introduction is  very good

-It would be great if we added the following :Types of Outcome Measures

          Primary outcomes

          Global State

         Mental state

         Quality of life

         Behavior

-inclusion and Exclusion Criteria : very clear

-Search Strategy: very clear

-Data Extraction : Well explained

-Data Synthesis and Meta-analysis : Well built

-Strengths and Limitations:  the past ten years (2012-2022)  is another Limitation .

Author Response

Dear Reviewer

Thank you for the valuable comments and suggestions which helped improve the manuscript.

Warm regards

Reviewer 3 Report (New Reviewer)

This study is a systematic review aims at examining ten years psycho-educational intervention programmes developed to address the psychological risks factors associated with non- 17 communicable diseases among adolescents, and to assess the effectiveness of the psychoeducational intervention programmes. I think the study is very importance in particular for these findings. In fact, this study provide useful information about the most useful tools to manage psychological risks factors associated to non-communicable diseases and for provide important information in term of intervention in these cases. I appreciated the detailed way by which the authors describe the introductory paragraph and the study methodology. However, I think the study could gain more strength by expanding the description of study’s objectives, hypotheses, findings and practical implications which are a lot. Below are reported specific comments about minor issues.

INTRODUCTION

11)     I appreciate the clear way by which the authors described the risks factor associated to adolescents’ health. However, my suggestion, is to further describe them by explicating the adolescents’ reasons and developmental challenges associated to this peak in risky behaviors for their health.

22)     I would also appreciate if the long introduction could be divided into sub-sections in order to make the reader more understandable and focalized.

33)     Finally, for what concerns the introduction, I would like to read a paragraph entitle “The present study”, containing the study overall aim and research question, the state of the art, the specific objectives and the study’s hypotheses.

MATERIALS AND METHODS

4)     I would like to read a paragraph that summarize the study’s findings.

STRENGHTS AND LIMITATIONS

54)     I suggest to better describe the study’s strengths and limitations.

CONCLUSION

65)     Please, describe in a more detailed way the study’s practical implications.

Author Response

Dear reviewer

Thank you for your valuable comments and suggestions that assisted in improving the quality of the manuscript.

Warm regards

This manuscript is a resubmission of an earlier submission. The following is a list of the peer review reports and author responses from that submission.

Round 1

Reviewer 1 Report

Attached

Author Response

Thank you for wonderful feedback, I hope the corrections are at your satisfactory.

Reviewer 2 Report

-Usually, In The Protocol, The Introduction The Introduction Is Not Long. It Would Be Better If The Introduction Was Shortened, And There Is A Lot Of Repeated Information

-Why It Is Important To Do This Review :It Put It In A Separate Paragraph

-O B J E C T I V E S:In A Special Paragraph

Types Of Studies 

I Think That The Types Of The Selected Studies Is Not Homogeneous :Which Include Observational Studies, Cross-Sectional Studies, Cohort Studies, Case-

Control Studies, And Randomized Controlled Trials (Rcts)

-We Cannot Compare A Observational Studies    With A Randomized Controlled Trials (Rcts ) So It Must Be Justified

-Types Of Interventions - Observational Studies, Cross-Sectional Studies :The Types Of These Studies Is Not  Interventions  So How Will They Be Included

-The Authors Did Not Mention How And What Criteria To Obtain  Types Of Outcome Measures

     -Primary Outcomes

    - Secondary Outcomes

- the review is very good

Author Response

Thank you for the wonderful feedback and I hope the corrections are at your satisfactory. 

Reviewer 3 Report

Review report : 

This protocol for ‘A Systematic Review Protocol for the effectiveness of psycho- 2 educational intervention programmes in addressing the psy- 3 chological risk factors associated with non-communicable dis- 4 eases among adolescents’ is well designed and enough to get interests from the clinicians and researchers in this field.

However, this protocol should be considered and reorganized due to some weak points.

1)     It is too broad and not clear descriptions aobut PICO.

In P: what doses it mean or defines about psychological challenges? How do the authors determine the population 10-19 years? Some country’s definition of adolescents is different.

In I : Psycho-education, CBT, Education are different contents and purpose. What is Intervention of this protocol? How can the authors make separation?

In C: What is comparator?

In O : The outcomes are more vague.

2)     How can the authors analysis by meta method? It needs to describe more precisely.

Author Response

(The authors gave the same response as above.)

Round 2

Reviewer 1 Report

Although the authors have improved some aspects, such as the introduction and the specificity of the search for studies, I consider that the data provided do not refute the results obtained. What are the results?

In addition, there are still errors in the font and in the structuring of the journal template. I recommend that everything should be written in the same style.

If the aim is to determine the different types of psychoeducational intervention programs developed to address psychological risk factors associated with non-communicable diseases among adolescents, the results of the study and comparisons should be presented. There is no clear debate among protocols, nor is there a consensus.

I believe that this study should not be published because of its flawed approach and lack of specificity.

Reviewer 3 Report

I think the authors have done a reat job summarizing and response each point in reviewing. Thanks. 

Author Response

Thank you for the positive response